# Experimental Analysis of Ductile Cutting Regime in Face Milling of Sintered Silicon Carbide

**DOI:** 10.3390/ma15072409

**Published:** 2022-03-24

**Authors:** Marvin Groeb, Lorenz Hagelüken, Johann Groeb, Wolfgang Ensinger

**Affiliations:** 1Kern Microtechnik GmbH, 82438 Eschenlohe, Germany; 2Department of Materials Science, Technical University of Darmstadt, 64287 Darmstadt, Germany; 3Microsystems Laboratory (LMIS1), Ecole Polytechnique Federale de Lausanne (EPFL), 1015 Lausanne, Switzerland; lorenz.hagelueken@epfl.ch; 4Independent Researcher, 64295 Darmstadt, Germany; groeb.johann@t-online.de

**Keywords:** ductile cutting regime, machining, silicon carbide, face milling, PCD tooling

## Abstract

In this study, sintered silicon carbide is machined on a high-precision milling machine with a high-speed spindle, closed-loop linear drives and friction-free micro gap hydrostatics. A series of experiments was undertaken varying the relevant process parameters such as feedrate, cutting speed and chip thickness. For this, the milled surfaces are characterized in a process via an acoustic emission sensor. The milled surfaces were analyzed via confocal laser scanning microscopy and the ISO 25178 areal surface quality parameters such as Sa, Sq and Smr are determined. Moreover, scanning electron microscopy was used to qualitatively characterize the surfaces, but also to identify sub-surface damages such as grooves, breakouts and pitting. Raman laser spectroscopy is used to identify possible amorphization and changes to crystal structure. We used grazing incidence XRD to analyze the crystallographic structure and scanning acoustic microscopy to analyze sub-surface damages. A polycrystalline diamond tool was able to produce superior surfaces compared to diamond grinding with an areal surface roughness Sa of below 100 nm in a very competitive time frame. The finished surface exhibits a high gloss and reflectance. It can be seen that chip thickness and cutting speed have a major influence on the resulting surface quality. The undamaged surface in combination with a small median chip thickness is indicative of a ductile cutting regime.

## 1. Introduction

Materials science and new materials are proving to be the key technology of the 21st century. Many processes can be conducted more efficiently and therefore saving resources if the environment is characterized by high thermal, chemical and mechanical loads. Traditional materials such as steel can even approach their theoretical limits in modern applications.

Technical ceramics are characterized by high hardness, large Young’s modulus, good thermal stability and inertness to chemical exposure [1,2]. A very hard technical ceramic, silicon carbide (SiC) is therefore predestined for parts in medical, (aero-) space, semiconductor and chemical uses [3]. Recent advances in fusion energy research highlight the importance of materials with low activation, where SiC is a prominent choice [4,5]. Producing parts from silicon carbide instead of soft metals brings forth several challenges in achieving both a good finish as well as tight tolerances [6].

SiC is classified as a hard and brittle material, which stems from the short sp ${}^{3}$-sp ${}^{3}$ bond length between the atoms [1]. Several key characteristics of SiC make the mechanical finishing difficult. Because of a very unfortunate ratio between the Young’s modulus and the hardness of SiC, a low ratio between tensile strength and shear strength, but also the combination of low density, dislocation mobility and low surface energy, it is considered very brittle. The higher portion of covalent bonding in SiC compared to ionic bonding (9:1) make it insensitive to high temperature deformations [7,8].

SiC displays a one dimensional polymorphism. All crystallographic structures are stacked tetrahedral arrangements of Si and C atoms with differing sequences. The used sintered polycrystalline silicon carbide in this paper is a hexagonal (6H) type. Figure 1 shows an example lattice cell.

Currently, high quality finishes are achieved via either chemical–mechanical polishing [9,10,11], lapping [12,13] or diamond turning [14,15]. Single point diamond turning (SPDT) is the predominant finishing process in ultra precision applications. It can generate surfaces with a very high form accuracy as well as single digit nanometric surface roughnesses. The price and availability of these machines has drastically improved [16,17,18,19,20].

Unfortunately, SPDT inherently shows low productivity and high part cost, [21] as well as limited shape freedom due to the accessibility from the diamond tool.

Finishing a part by hand is undesirable, as manual finishing leads to form errors, but also increases the cost. Ultra precision machining (UPM) can produce parts with the associated final specifications without any manual finishing, and thus reduce the cost [22]. Nevertheless, UPM machines show a low productivity, and often cannot provide the full operations such as roughing and semi finishing of a part on one machine. A second costly CNC machine needs to be added, increasing complexity and cost. The overall efficiency of a UPM finishing machine is considered low [21].

High-quality surface finishing with monocrystalline diamond tools has been possible in a number of brittle materials such as Germanium [23], glass [24], tungsten–carbide [25], but also softer non-ferrous material, for instance aluminum [26].

Since the late 1980s, SPDT uses the phenomenon of a brittle-to-ductile transition [27,28] to achieve high quality finishes in brittle materials. The effect of brittle materials behaving in a ductile regime has first been recorded by King et al. in 1954 [29], who discovered that rock salt exhibits plastic deformation under very-high loads. Further the early research was undertaken by Hertzfeld, who showed a high pressure phase transformation of Germanium in ductile milling [6,30,31,32,33]. Recent research has focused heavily on ductile regime grinding of brittle materials [30,34,35].

This paper explores the fabrication of high-quality surfaces with a geometrically defined cutting edge in face milling (an interrupted cut with a rotating tool) of sintered SiC.

## 2. Materials and Methods

The used SiC is a sintered commercial grade produced by FCT Ingenieurkeramik GmbH, Germany. Table 1 sums up the composition as well as general properties. The material was ground flat and round after sintering on a conventional diamond grinding machine by the manufacturer.

In order to explore the parameters towards achieving a ductile cut, but also understand the effect on the material, a series of experiments are performed.

A first experiment series varying the cutting speed;A second experiment series exploring the influence of the feedrate on achieving ductile behavior;A third, orthogonal matrix based experiment series analyzing the correlation between the first two experiment series;A number of scratch tests, determining a critical transition depth from brittle to ductile behavior;The production of 3 sample surfaces (diamond ground, milled in a non-ductile and a ductile cutting regime) for further analytical purposes such as XRD and SAM.

All experiments were performed on a Kern Micro HD. It is a highly precise vertical CNC machining center. The drive system is a friction free microgap hydrostatic design, driven by linear motors. The position feedback loop is controlled by Heidenhain glass scales (resolution: 7 nm), which in combination with a full spatial geometric compensation enable positioning errors in the sub micrometer range. The milling spindle (manufacturer: Fischer) features an HSK 40 interface and a maximum rotational speed of 42,000 min${}^{-1}$ with a maximum power of 15.6 kW.

The material used is an industry standard grade sintered SiC. Roughing and semi finishing was performed with a 2 mm polycrystalline diamond (PCD) tool manufactured by 6C Tools AG, Switzerland. A second, identical tool was used for finishing. The tools were held in a Regofix PowRgrip toolholder. The sample specimen was fixed in a low profile vise (Lang 125 mm) in steel soft jaws. The used tool paths are written directly on the control, a Heidenhain TNC 640. The machines allowed contour deviation was set to 0.0005 mm. All experiments were performed under flood coolant (Oelheld SintoGrind TC-X 1500).

During the experiments, extra care was taken to reduce extraneous influences such as thermal expansion and drift. For this, a certain spindle warm up was performed, and the environment temperature stabilized via air conditioning. The workpiece was tempered with the machine coolant for a period of 10 min at the beginning of every experimental series. During the experiments, acoustic emission (AE) signals were recorded. For this, a broadband (600 kHz) magnetic sensor was connected to the steel vise and recorded via a Marposs AE6000 amplifier connected to a Picoscope 5244D digital oscilloscope. Figure 2 shows the experimental setup. After machining, the samples were cleaned in an ultrasonic bath filled with isopropanol alcohol. The quantitative surface measurements according to ISO 25178 were taken on a Confovis DuoVario confocal laser scanning microscope (CLSM), where measurements show an uncertainty of +/−3 nm. Topographic pictures of the surface for qualitative analysis were taken on a Thermo Fischer Phenom XL scanning electron microscope (SEM). Raman laser spectroscopy (Renishaw inVia, wavelength 532 nm) was undertaken on selected surfaces. For every analyzed field, 6 accumulations at 15 s acquisition time were taken. The samples produced in the fifth experiment series were then analyzed via grazing incidence x-ray diffraction (XRD) measurements in a Bruker D8 diffractometer. An acquisition time of 15 s was selected. The angle (2 θ) was recorded from 10° to 60°, with increments of 0.03°. Scanning acoustic microscopy (SAM) was performed at 80 and 400 kHz on a PVA-Tepla microscope.

In general, a ductile cutting regime is reached by lowering the chip thickness towards a critical value where a brittle-to-ductile transition happens. This critical chip thickness is in the range of single digit to several tens of nm, depending on the tool geometry, material and experiment. This is achieved by using very-precise single point diamond turning ultra-precision machines. In milling, it is implausible to achieve a dimensional load in the manometer range, as both the machine tool repeatability as well as the tool run out are often orders of magnitude larger. Because of the way chips are formed with an interrupted cut, the chip thickness (in climb milling) starts at a maximum value (the feed per tooth) and rapidly diminishes in thickness, especially with lower radial engagements, a phenomenon called chip thinning. Figure 3 shows the schematic geometrical definition of the median chip thickness Hm.

The chip thickness can be calculated by taking the segment height of a circle at the middle between the upper and lower engagement angle. Equation (Equation 1) describes this relationship for an endmill. Here, $\varphi_{s}$ is the radial engagement angle, $a_{e}$ the radial engagement, ι the angulation, *D* the endmill diameter and F_z_ the feed per tooth.
(1)$$H_{m}=\frac{360{}^{\circ}}{\pi \cdot \varphi_{s}}\cdot \frac{a_{e}}{D}\cdot F_{z}\cdot sin\left(\iota \right)$$

Because the radial engagement has a major influence not only on the median chip thickness, but also on the mechanical and therefore the thermal load of the tool, the toolpath consists of evenly offset passes, that feature a constant radial engagement. Figure 2b schematically sketches the toolpath kinematics. All measurements were taken at a region, where the tool is fully stabilized over the sample specimen, to enable comparable results.

Before the experiments, the surface was faced with a larger, 6 mm PCD endmill. An axial stock of 0.1 mm was left, thus defining the cutting depth for the experiment series. A cutting speed of 500 m/min with a feedrate of 300 mm/min and both a cutting depth and radial engagement of 0.1 mm was used. The semi finishing operation was conducted with flood coolant. Before the facing operation, a spindle warm up period of 300 s was performed.

The first experiment series explores the influence of the cutting speed on the cutting process. The experiment was conducted with a 2 mm diameter bull nose endmill, featuring a corner radius of 0.05 mm. The tool features 15 cutting edges with an edge sharpness between 1–2 µm. A short break in period of 1 min is undertaken to stabilize initial tool wear before the experiment series. Every 3 parameter sets, the tool was visually inspected at 100 times magnification for tool wear. A common width of cut (radial engagement) of 0.1 mm was used in this experiment. After every rotational speed change, the spindle was warmed up for a period of 300 s to reduce variation through thermal elongation. Table 2 shows the experimental parameters. Because the feedrate was kept at a constant rate, the median chip thickness decreases.

In order to have comparable results, a new endmill was used for the second experiment series. An identical break in period of 1 minute was undertaken, and all cutting parameters (depth of cut, width of cut, warm-up periods, coolant) were adopted from the first experiment series. Table 3 shows the variation of the feedrate at a fixed cutting speed, thus analyzing the influence of the median chip thickness on the cutting process.

The third experiment series looks at complex interaction between cutting speed and chip thickness. For this, the feedrate is calculated based on 4 different levels of chip thickness and cutting speed. Table 4 lists the cutting parameters of the third experiment series. The tool was periodically checked for wear. Identical thermal warm-up routines as in the first two experiment series were applied.

The material blank (see Figure 4) is then cleaned in an ultrasonic cleaner with isopropanol, before going through a commercial hot ethanol vacuum cleaner. Afterwards, the experiment parameter fields were analyzed in the scanning electron microscope (acceleration: 5 kV, detector: SE) for their qualitative surface composition, in the CLSM for the quantitative surface parameters and selected fields in a Raman laser spectrometer for their microstructure.

Results from the first three experiment series prompted two trials on determining the brittle-to-ductile transition critical chip thickness. First, a rotating ball endmill was ramped into the material at a very shallow angle (0.001 mrad). Afterwards, the workpiece was held perpendicular to the cutting edge, and the tool was again ramped at a shallow angle into the material. The resulting scratch marks where analyzed in both the scanning electron microscope as well as the CLSM. From the step height, a critical thickness was derived.

Afterwards, a number of smaller samples (see Table 5) were produced to explore the material properties of the different cutting modes. For this, a reference (diamond ground) sample, a milled sample with high median chip thickness (brittle material removal mode) and a milled sample with low median chip thickness (ductile cutting mode) were produced. The cutting depth was fixed at 0.02 mm, to enable a comparable result between the diamond grinding bit and the milling tools.

## 3. Results

### 3.1. First Experiment Series

The first experiment series produced smooth, shiny but not mirror-like surfaces without visible tool marks or breakouts. The measured surface roughness Sa /Sq is in the range of 100 to 270 nm (see Figure 5). It can be seen that with diminishing median chip thickness (Hm), the roughness slightly improves. The surface roughness, while being an industry standard parameter, does not characterize the surface in a major way. For this, the surface bearing parameter (Smr) was evaluated as well. In sintered ceramics, where the surface formation usually is dominated by brittle breakouts as well as pulled out grains, the roughness cannot display the composition of the surface. The parameter Smr meanwhile takes into account the amount of undamaged surface, as it displays the material ratio at a certain height (0.5 µm in this study), and thus shows a significant drop (exp. 15–18) when the cutting mechanism changes from brittle failure to ductile mode (see Figure 5).

Taking a further look at the ISO 25178 parameters, it can be seen that both the areal material ratio (Smr) as well as the peak extreme height (Sxp) are not influenced by the median chip thickness (see Figure 6). The texture aspect ratio (Str), characterizing the isotropy of the surface, increases with a median chip thickness below 80 nm.

These results match up with the qualitative surface, easily visible on SEM micrographs. The third experiment parameter (Hm = 187 nm) shows a surface dominated by brittle failure and granular tear-outs (see Figure 7a). The parameter set 17, characterized by a low Hm (56 nm) shows large areas with no tear-outs and homogeneous surface, exemplary of an inter-crystalline cut and therefore pointing towards a ductile cutting regime.

The recorded AE signals are transformed via a fast Fourier transformation (FFT) (method: welch, window: 2048) into a power spectral density (PSD). By taking the integral of the PSD, a normalized PSD sum could be created for every experiment. Figure 8 shows the relation between Smr and the PSD sum.

It can be seen that on parameter sets with lower surface damages (meaning a lower Smr value, i.e., exp. 16–18), the PSD sum is significantly higher than on those parameter sets with large amounts of brittle failures. The high pressure phase transformation requires additional energy compared to the brittle fracture, and could thus explain this rise in PSD sum values. A relation between other surface parameters such as Sa, Sq, could not be undertaken by the AE PSD. Looking at the amplitude of the PSD signal, a closer correlation (Figure 9) with the surface roughness Sq can be seen. The correlation is especially strong at higher cutting speeds, and in conjunction with the qualitative appearance of those parameter sets suggests that the surface roughness at higher cutting speeds is dominated not by brittle failure of the surface composition, but vibrations of the cutting tool imprinting on the surface.

The parameter set 17 (cutting speed 250 m/min, median chip thickness 56 nm) showed especially intact surfaces. As the brittle-to-ductile cutting mode transition is dependent on a critical chip thickness, and often is in the range of lower double digit manometer values, this could point towards a successful ductile cutting regime in this parameter set.

A recovered chip is shown in Figure 10. To verify that it is indeed a SiC chip, an EDS analysis was undertaken. At an acceleration voltage of 5 kV, the EDS shows only silicon (atomic concentration 23.6 %) and carbon (atomic concentration 76.4 %). The higher carbon content can be explained by the relatively low vacuum strength of the used SEM, thus having a surplus of atmosphere inside.

### 3.2. Second Experiment Series

The second experiment series, which kept cutting speed constant while increasing the feedrate, thus exploring the influence of median chip thickness on the result, also produced highly reflective surfaces. Figure 11 shows the surface roughness and surface bearing parameter Smr in relation to the median chip thickness. It can be seen that at very low median chip thickness (Parameter set 1, median chip thickness 22 nm) the surface roughness is very high, dropping significantly at the next parameter set. This suggests that there is a minimum chip thickness, below which the tool is only introducing surface errors, similar to how a tool would rub in metal once the feed per tooth drops below a certain threshold. No noteworthy correlation at further surface parameters or the AE signals can be seen.

### 3.3. Third Experiment Series

The third experiment series changed both median chip thickness and cutting speed along several different levels, thus the interaction between these parameters was explored. Figure 12 shows the roughness in dependence of the median chip thickness. No clear correlation can be seen. An analysis of variance (ANOVA) looking at the interaction between all quality parameters and the median chip thickness, the cutting speed but also the interaction between these two showed no significant influences.

Two remarkable parameter sets are shown as SEM micrographs in Figure 13. The left side is dominated by excessively deep brittle fractures and was produced at a median chip thickness of 67 nm, with a cutting speed of 150 m/min. Figure 13b shows the same median chip thickness, but at a cutting speed of 200 m/min. The large difference in intact surface is indicative of the difference between a brittle (failure mode) material removal and a ductile cutting regime.

It can be concluded that on certain parameter sets, especially with Hm below 80 nm, indications towards a ductile cutting regime exist. The cutting speed has a modest influence on the result, and generally improves the results in quality, as long as the vibration of the tool is not noticeable. Furthermore, trials into the cutting depth have shown no influence on the surface composition, but are not shown here for the sake of brevity. Acoustic emission spectroscopy has shown promise in supervising the process both for the resulting roughness, but also through a sharp increase in the PSD sum on certain parameter sets.

### 3.4. Fourth Experiment Series—Determination of Brittle to Ductile Transition Depth

In order to determine the exact critical chip thickness at which a brittle-to-ductile transition happens, a scratch test was performed. For this, the tool was ramped at a shallow angle (0.002°) into the material. Figure 14 shows the region of interest. The tool first cuts in a ductile cutting regime (region c), whereas at a certain depth, a transition towards brittle material removal (region a) comes to pass. The brittle removal is characterized by brittle breakouts of individual grains along the cutting path (region b). By analyzing the height with a CLSM, the critical depth can be determined. A transition depth of 55 nm was found in three separate scratch tests. Afterwards, the tool was oriented so that the cutting edge is perpendicular to the cutting surface and dragged along the surface. A transition depth of 46 nm was calculated.

### 3.5. Fifth Experiment Series—Baseline Comparison

The first three experiment series explored the process of milling with a geometric defined cutting edge, whereas the fourth experiment found a critical transition depth. With this knowledge, three samples were produced and then subjected to scanning accoustic microscopy. Figure 15 shows the resulting micrographs. By applying a cut-off filter, certain depths can be pictured.

It can be seen that the diamond ground sample has a large amount of sub-surface damages even at 120 µm depth, whereas the brittle-milled sample has fewer damages. The ductile-milled sample shows no SSD at 120 µm depth and very little at 90 µm. By increasing the frequency of the ultrasonic emitter, the interaction depth decreases while the spatial resolution increases. Figure 16 shows the surface composition of the brittle-milled and ductile-milled samples at 400 kHz.

While the wavelength of the ultrasonic wave is still quite large for a modern analytical process, the amplitude of the wave is very small. This means, that when the wave enters a crack in the material, there is no reflection and therefore a black pixel appears, even though the detected crack width might be smaller than the spatial resolution of the SAM. The brittle-milled sample (Figure 16a) shows these fissures all over the surface, which are not detectable in SEM and CLSM analysis. Because of the polycrystalline nature of the material, they are likely self-closed cracks.

The ductile-milled sample has no apparent surface damages.

### 3.6. Compositional and Crystallographic Analysis

The previous chapters have shown parameters with a remarkable surface quality both in terms of roughness as well as the amount of sub-surface damages. Aside from the physical appearance of the material, changes could theoretically happen in both the chemistry as well as the micro structure (e.g., amorphization, change of the crystal structure). For this reason, Raman laser spectroscopy was conducted on certain sample fields from the first experiment series, but also grazing incidence XRD on the samples produced in the fifth experiment series.

Raman spectra were baseline corrected and normalized to the 969 cm${}^{-1}$ peak. The resulting plots for eight sample fields are shown in Figure 17. They exhibit the (poly)crystalline 6H-SiC transverse optic (TO) modes at 788 cm^−1^ as well as longitudinal optic (LO) mode 969 cm^−1^. Lorentzian peak fitting was performed in order to determine peak intensity, center position and FWHM values among which the most important ones are summarized in Table 6.

Due to the unit cell c- and a-axis correlation (see Figure 1) to the LO and TO mode, respectively, their ratios (LO/TO) are a measure for the SiC grain orientation [36,37] which is random in the case of polycrystalline SiC. The peaks at 797 cm${}^{-1}$ are associated with 3C- and 4H-SiC polytypes and hence serve as an indicator for the presence of stacking faults (SF) in 6H-SiC. A higher TO_SF_/TO ratio (given in Table 6) therefore suggests more machining-induced defects in the respective material. Furthermore, a shift of the TO and LO peak positions may indicate residual stress in the excited material. Values calculated for LO according to the model of Liu and Vohra [38] are listed in Table 6. These calculated values are best for the REF field whereas V1-01, V3-01 and V3-16 exhibit the highest signs of SF generation and residual stress. In all eight sets of parameters, no machining-induced amorphization occurs, which would be indicated by a significant broadening of the Raman spectra [6].

The grazing incidence XRD analysis was conducted on a Bruker D8 diffractometer. Figure 18 shows the recorded spectra. It can be seen that SiC in the 6H structure (dashed, vertical lines) is the dominating reflex. The recorded low intensity peaks at other angles are most likely impurities in the sintered technical ceramics, for example the reflex at 26.59° could match silicon dioxide (SiO_2_). The three samples show no discernible deviations, and can thus be considered identical in their crystallographic structure.

## 4. Discussion

### 4.1. Brittle-to-Ductile Transition

The first three experiments explored influencing factors towards a ductile cutting regime. It can be seen that below 80 nm, the surface bearing parameter Smr drastically changes, and at higher cutting speeds, the surface roughness improves. The qualitative surface shows clear intracrystalline cutting, indicative of a ductile cutting regime. A recovered chip is not the expected broken up debris, but a twisted, metal-like chip, proving the ductile cutting regime. It has been reported that under sufficiently high pressures, 6H-SiC can transform towards different phases [39,40]. During these experiments, the produced chips were milled in a climb direction, meaning that the chip starts at the thickest point (equal to the feed per tooth), and rapidly grows thinner, a phenomenon described in Section 2. At a certain point, the cutting edge radius is larger than the remaining chip thickness, thus creating an instantaneous, highly negative rake angle. This rake angle induces a high amount of hydrostatic pressure into the material.

Studies have shown that in SPDT, a ductile cutting regime in SiC is achievable, usually at cutting depths of less than 30 nm [6,15,31]. The presented parameters in this study show a much larger depth of cut (0.1 mm), but also a ductile behavior up until 50–60 nm. A likely reason here is the much higher cutting speed in milling instead of diamond turning. The cutting speeds achieved in this study surpass those easily achievable on UPM machines by one order of magnitude. Studies have shown that both temperature and deformation speed have a large impact on achieving a phase transformation in SiC [41,42]. Gilman has shown [43] that the addition of shear stress onto hydrostatic stress lowers the pressure requirements for a phase transformation by a large amount. The sheared lattice can, on neighboring cells, close the energy band gap and thus become metal-like, enabling dislocation mobility. Figure 19 shows this shear stress effect onto a cubic lattice. This is further supported by the scratch tests performed in this study. It can be seen that a significantly higher ductile transition depth (55 instead of 46 nm) can be achieved by rotating the tool at a higher frequency, achieving a higher cutting speed than by just dragging it along the surface, where the movement speed equals the cutting speed.

During Raman laser spectroscopy, no significant change to the chemical composition or the phases could be detected. Moreover, grazing incidence XRD showed that both the ductile and the brittle samples have the original 6H structure of SiC. This means that the high pressure phase is metastable, and reverses back towards the original structure once the cutter exits the material. It has been shown that the reversal towards the original structure is dependent on the rate of decompression [44], which could potentially explain why in milling the amount of amorphization is lower than in SPDT.

### 4.2. Milling Kinematics towards a Partial Brittle to Ductile Transition

The amount of detected sub-surface damage (see Section 3.5) is much lower during ductile cutting, even though the material experiences a large amount of hydrostatic stress. Moreover, the remaining tensile stress in the material is also lower in parameter sets that show ductile behavior. The chip geometry during milling is more complex than in SPDT. Figure 20 shows a schematic of the brittle-to-ductile transition. The chip formation starts at a thickness equal to the feed per tooth. This value is above the critical thickness for a brittle-to-ductile transition. At this point, the force vector points mostly in front of the tool, inducing sub-surface damages in direction of the feed. At a certain point, the median chip thickness Hm is below the critical thickness and a ductile cutting regime sets in. At this point, no further damages are induced into the material. After the tool exits the material, the hydrostatic pressure is relieved instantaneously and thus the material reverts back to the original structure.

Following this, the next tooth of the cutter enters the material, repeating the process. The induced micro-cracks were removed in the brittle cutting regime, before the ductile regime sets in again. This is supported by the lower amount of sub-surface damage in ductile mode (see Section 3.5).

## 5. Conclusions

In this study, a number of experiments exploring a possible ductile cutting regime in sintered SiC while face milling with a geometrically defined cutting edge were undertaken. It can be concluded that:The resulting surfaces were characterized not only for their roughness, but also for the ISO 25178 functional parameters. It can be seen that below a certain critical chip thickness of 55 nm, the areal material ratio improves spontaneously. A surface roughness Sa of 0.1 to 0.2 µm was achieved.SEM micrographs show a smoother, less broken up surface on low median chip thickness parameter sets.A recovered SiC chip shows a clear “twist-like“ shape.Scanning acoustic microscopy revealed a much reduced amount of sub-surface damages on a ductile-milled sample, compared to brittle-milled or ground samples.

The significant change in the surface parameters at low median chip thicknesses, combined with the recovered chip, reduced sub-surface damage, but also the qualitative appearance of the surfaces point towards a ductile cutting regime.

Compositional analysis showed that the surfaces generated do not change in their chemistry (RLS) or crystallographic structure (XRD), proving that no detectable amorphization takes place.A correlation between the acoustic emission data and the surface bearing parameter Smr shows an increase in the power spectral density when a ductile cutting regime is achieved. Further research is needed here, but a potential use could be in-process optimization and overview. A spontaneous change in the AE PSD could point towards either a tool failure, or brittle material removal mode.

Further research should look into the influence of the cutting tool geometry, especially cutting edge geometry, on the cutting process, as well as the influence of lubrication agents on the process.

## Figures and Tables

**Figure 1 materials-15-02409-f001:**
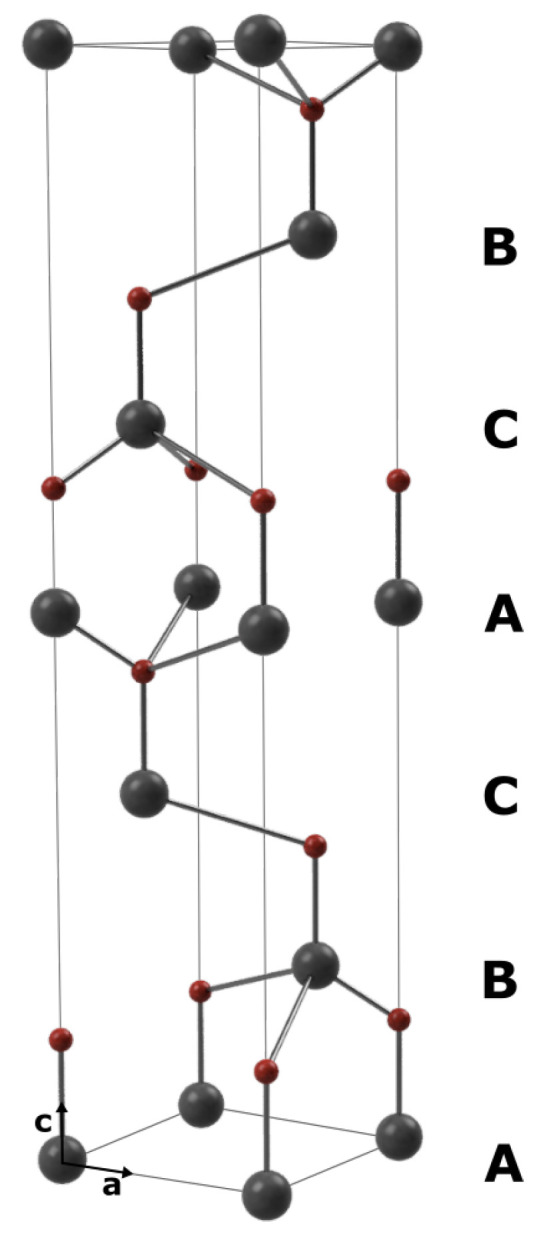
The 6H stacking sequence of SiC.

**Figure 2 materials-15-02409-f002:**
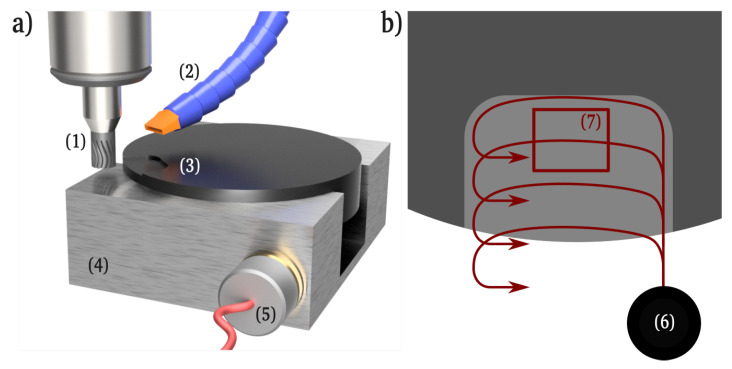
The experimental setup (**a**) showing the tool with toolholder (1), the coolant supply nozzle (2), sample specimen (3) held in soft jaws from steel (4), with the magnetic acoustic emission sensor (5) (**b**) the toolpath kinematics (6) showing the measured region of interest (7).

**Figure 3 materials-15-02409-f003:**
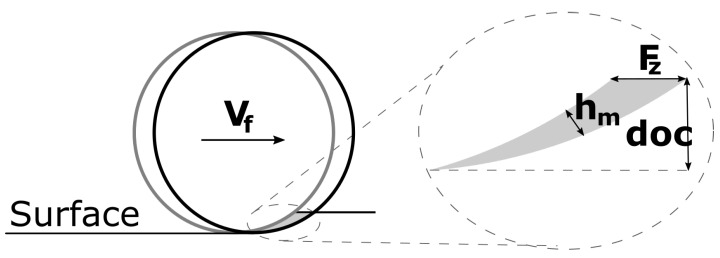
The geometrical definition of the median chip thickness Hm.

**Figure 4 materials-15-02409-f004:**
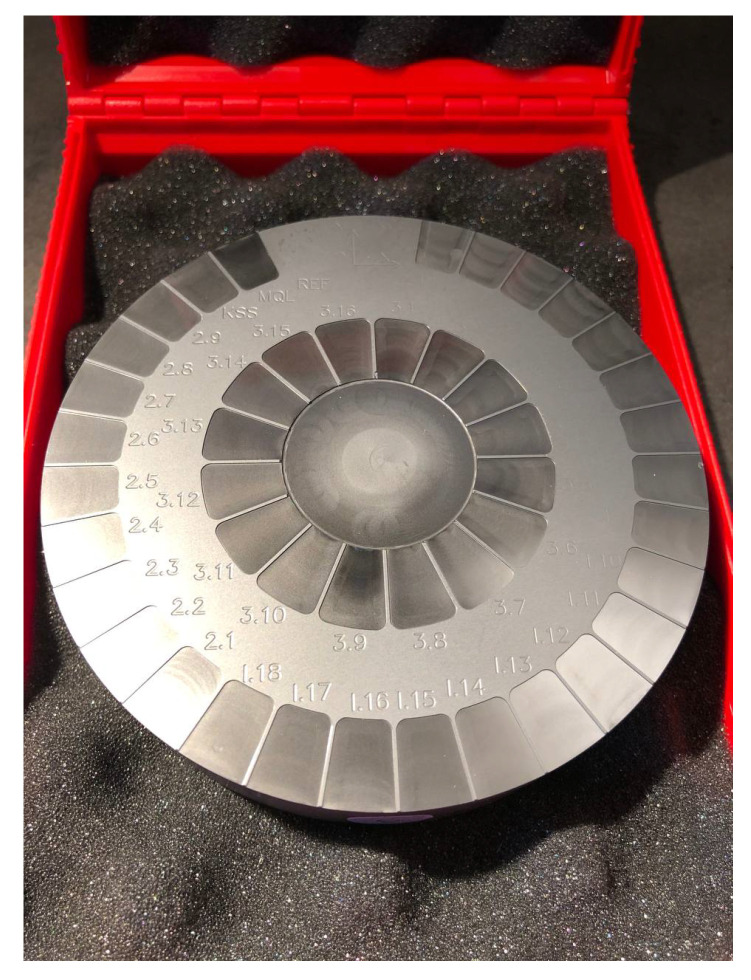
The finished SiC sample blank with all 3 experiment series on it. Diameter: 100 mm.

**Figure 5 materials-15-02409-f005:**
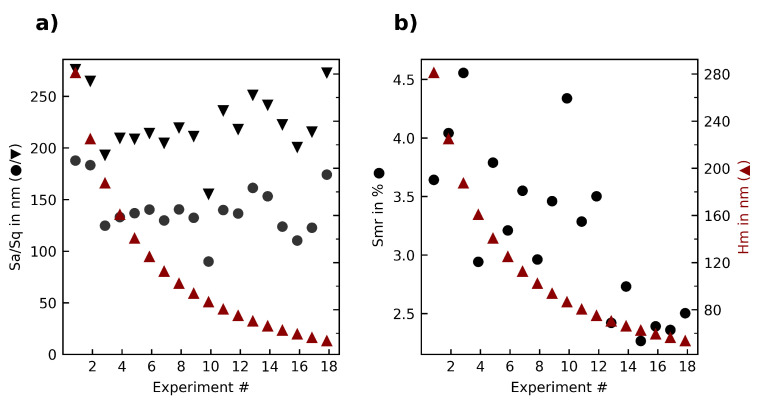
Overview of the (**a**) roughness parameters Sa, Sq; (**b**) surface bearing parameter Smr in relation to the median chip thickness Hm in the first experiment series.

**Figure 6 materials-15-02409-f006:**
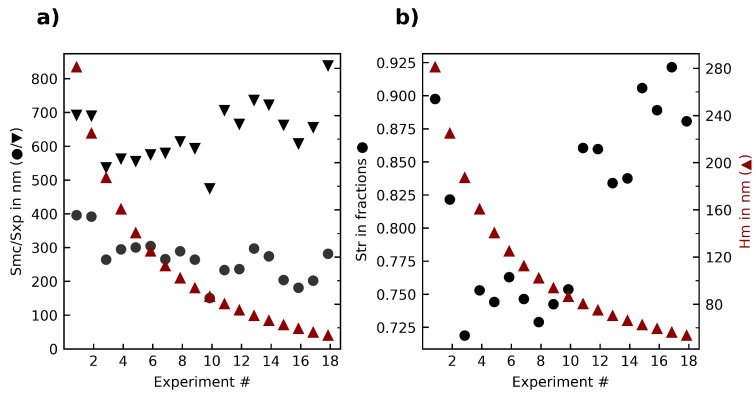
Overview of the (**a**) material ratio parameters Smc, Sxp; (**b**) texture aspect ratio (Str) in relation to the median chip thickness Hm in the first experiment series.

**Figure 7 materials-15-02409-f007:**
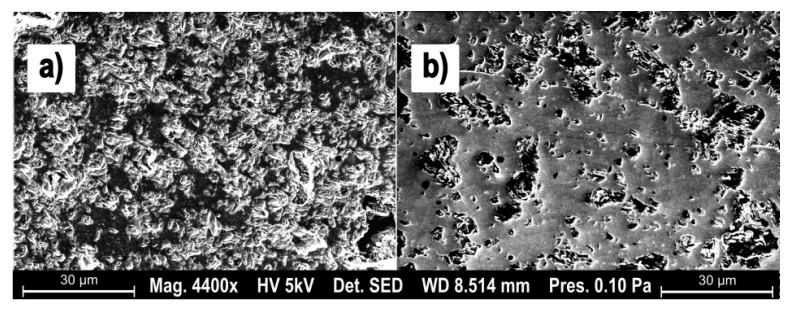
SEM micrographs showing two surfaces: (**a**) brittle material removal; (**b**) ductile material removal.

**Figure 8 materials-15-02409-f008:**
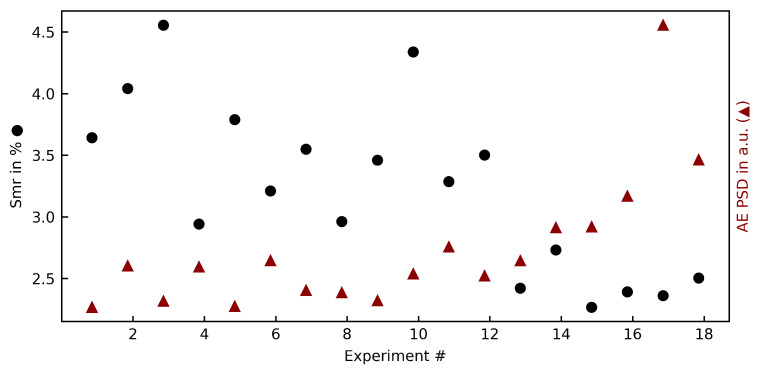
The integral of the AEPSD in relation to the surface bearing parameter Smr.

**Figure 9 materials-15-02409-f009:**
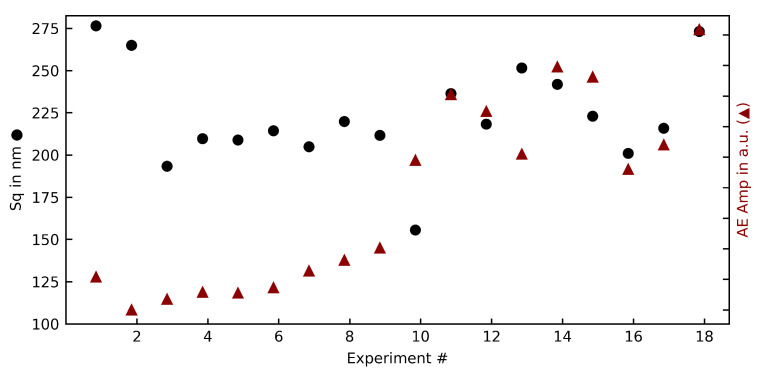
The AE amplitude in a.u. (average over 1 s) in relation to the quadratic surface roughness Sq.

**Figure 10 materials-15-02409-f010:**
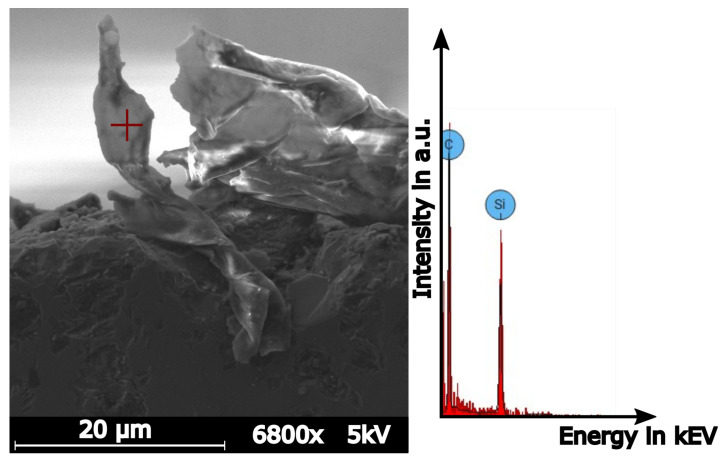
A recovered ductile chip. EDS analysis shows a composition of silicon and carbide.

**Figure 11 materials-15-02409-f011:**
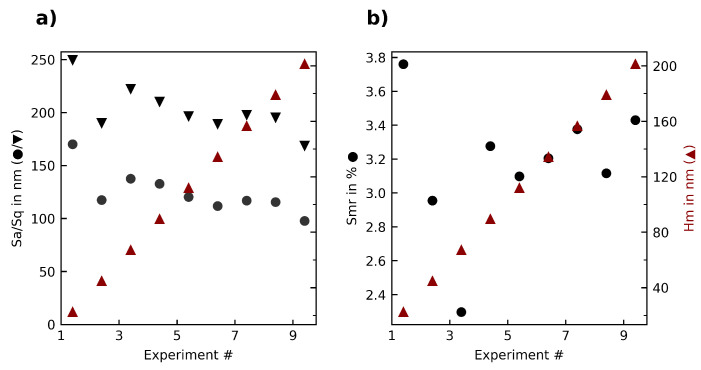
Overview of the (**a**) roughness parameters Sa, Sq; (**b**) surface bearing parameter Smr in relation to the median chip thickness Hm in the second experiment series.

**Figure 12 materials-15-02409-f012:**
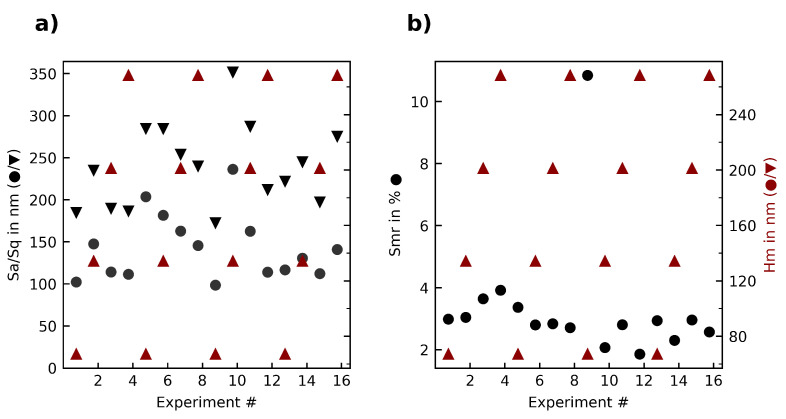
Overview of the (**a**) roughness parameters Sa, Sq; (**b**) surface bearing parameter Smr in relation to the median chip thickness Hm in the third experiment series.

**Figure 13 materials-15-02409-f013:**
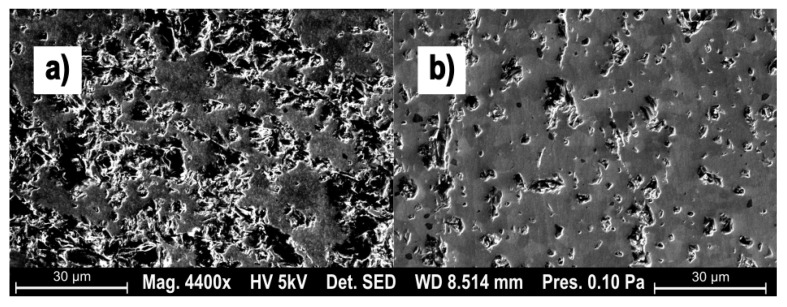
SEM micrographs showing two surfaces: (**a**) brittle material removal; (**b**) ductile material removal.

**Figure 14 materials-15-02409-f014:**
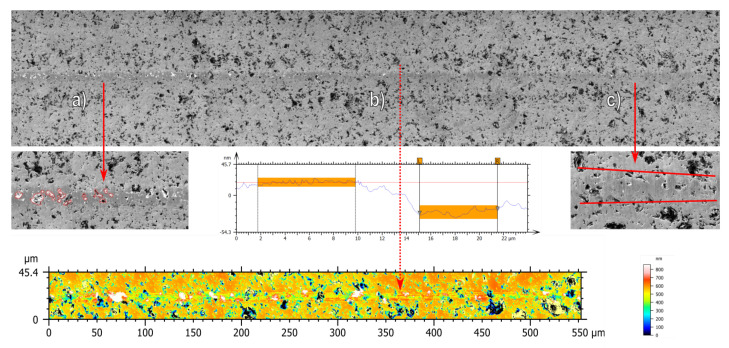
SEM and CLSM micrographs showing the determination of the brittle (**a**) to ductile (**c**) transition depth (**b**).

**Figure 15 materials-15-02409-f015:**
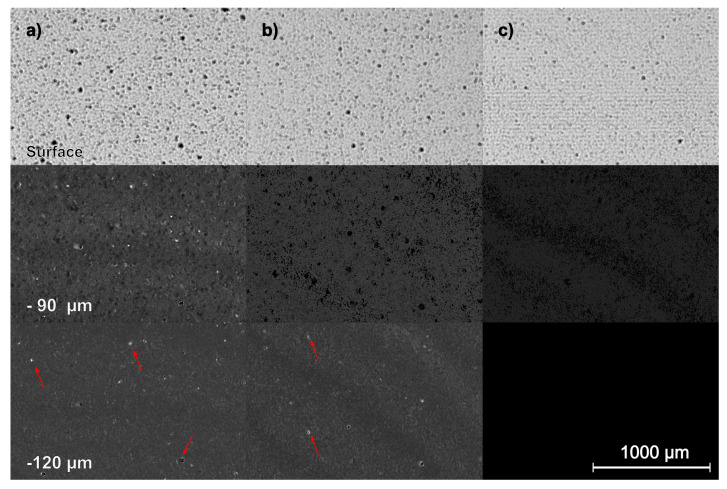
SAM micrographs showing the composition at the surface, a depth of −90 and −120 micrometer of (**a**) ground (**b**) brittle material removal milled (**c**) ductile cutting regime milled. Note: the micrographs below the surface layer are raised identically in their contrast and brightness to allow for easier identification of noteworthy defects.

**Figure 16 materials-15-02409-f016:**
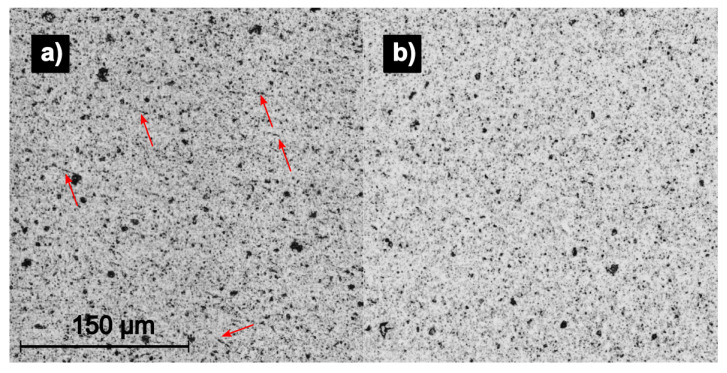
SAM micrographs showing the surface composition of: (**a**) brittle material removal milled; (**b**) ductile cutting regime milled. Red arrows point out self-closed cracks in the material.

**Figure 17 materials-15-02409-f017:**
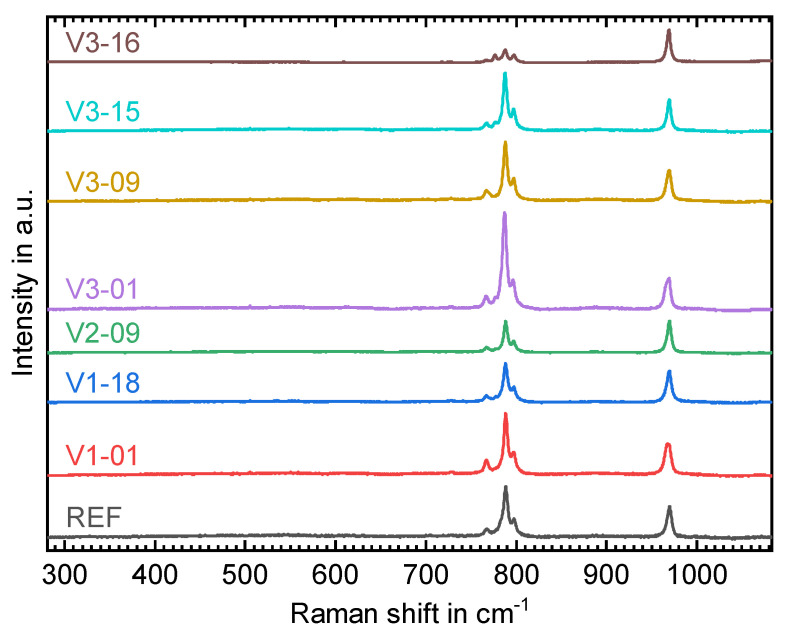
Raman spectra of eight selected sample fields of machined 6H-SiC.

**Figure 18 materials-15-02409-f018:**
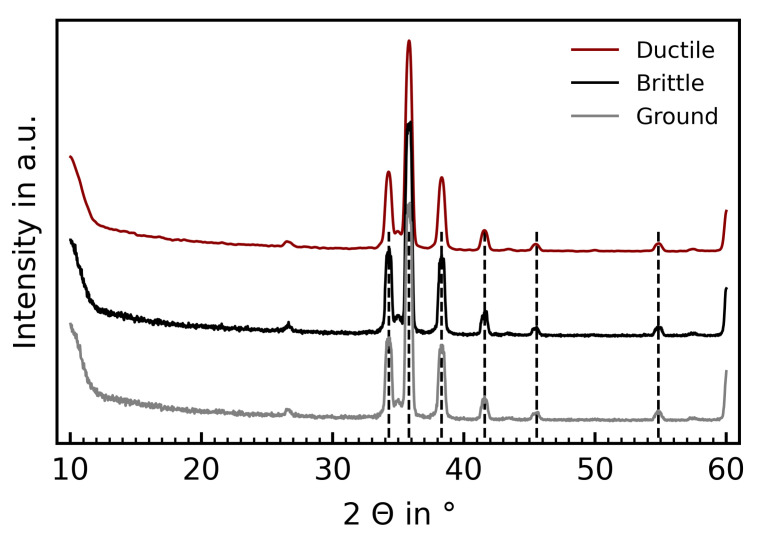
Recorded XRD spectra for the ductile-milled, brittle-milled and diamond ground sample surfaces. The dashed lines show the corresponding 6H structure reflexes (powder diffraction file 00-073-1663).

**Figure 19 materials-15-02409-f019:**
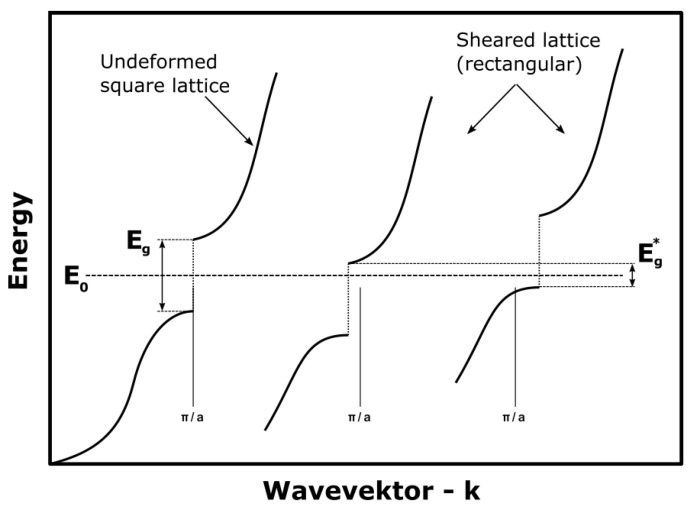
The influence of shear rate onto wave vectors. Shown is the energy vs. wavevector for a simple cubic lattice. Adapted from [43].

**Figure 20 materials-15-02409-f020:**
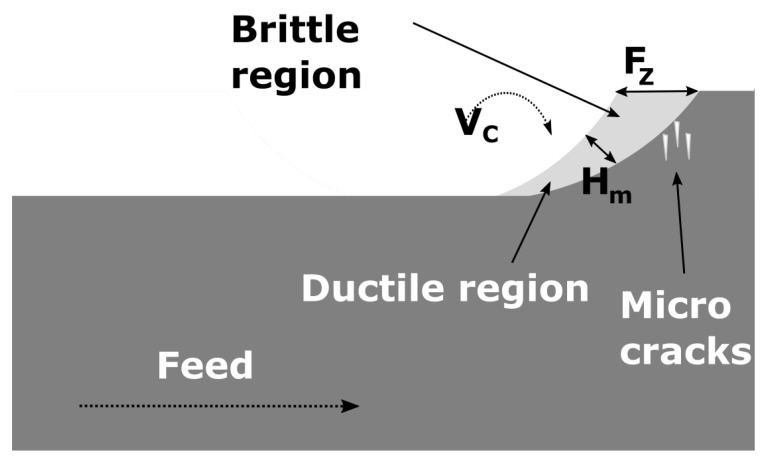
The kinematics showing the brittle to ductile transition during chip formation.

**Table 1 materials-15-02409-t001:** The composition and properties of the used sintered SiC.

Property	Unit	-
Composition	SiC (weight %)	>97
Composition	C/B_4_ (weight %)	<3
Density	g/cm${}^{3}$	>3.12
Porosity	%	<3
Hardness	GPa	22
Young’s modulus	GPa	400
Compressive strength	MPa	>3000
Flexural strength	MPa	450
Fracture toughness	MPa m${}^{1/2}$	3.00
Poisson’s number		0.16

**Table 2 materials-15-02409-t002:** Overview of the first experiment series.

Parameter	Cutting Speed	Feedrate	Med. Chip Thickness
Set	in m/min	in mm/min	in nm
1	50.0	150	281
2	62.5	150	225
3	75.0	150	187
4	87.5	150	161
5	100.0	150	140
6	112.5	150	125
7	125.0	150	112
8	137.5	150	102
9	150.0	150	94
10	162.5	150	86
11	175.0	150	80
12	187.5	150	75
13	200.0	150	70
14	212.5	150	66
15	225.0	150	62
16	237.5	150	59
17	250.0	150	56
18	262.5	150	54

**Table 3 materials-15-02409-t003:** Overview of the second experiment series.

Parameter	Cutting Speed	Feedrate	Med. Chip Thickness
set	in m/min	in mm/min	in nm
1	187.5	45	22
2	187.5	90	45
3	187.5	134	67
4	187.5	179	89
5	187.5	224	112
6	187.5	269	134
7	187.5	313	157
8	187.5	358	179
9	187.5	403	201

**Table 4 materials-15-02409-t004:** Overview of the third experiment series.

Parameter	Cutting Speed	Feedrate	Med. Chip Thickness
Set	in m/min	in mm/min	in nm
1	100	72	67
2	100	143	134
3	100	215	201
4	100	286	268
5	150	107	67
6	150	215	134
7	150	322	201
8	150	430	268
9	175	125	67
10	175	251	134
11	175	376	201
12	175	501	268
13	200	143	67
14	200	286	134
15	200	430	201
16	200	573	268

**Table 5 materials-15-02409-t005:** Overview of the produced 3 samples—ground, milled and milled in a ductile cutting regime.

Process	Tool	Cutting Speed in m/min	Feedrate in mm/min	Med. Chip Thickness in nm
Grinding	Grinding Cup D126	187.5	45	22
Milling, ductile	PCD, 2 mm	187.5	90	45
Milling, brittle	PCD, 2 mm	187.5	134	67

**Table 6 materials-15-02409-t006:** Key figures derived from Raman Lorentzian peak fitting, calculated ratios and derived residual stress.

	LO	LO	TO	TO	TO_SF_	LO/TO	TO_SF_/TO	LO	LO
#	Int.	Pos.	Int.	Pos.	Int.	Int. Ratio	Int. Ratio	Peak Shift	Res. Stress
	a.u.	cm${}^{-1}$	a.u.	cm${}^{-1}$	a.u.	%	%’	cm^−1^	GPa
REF	8271	969.5	14794	788.1	2606	56	18	0.6	−0.15
V1-01	11382	967.9	14723	788.3	4552	77	31	2.2	−0.57
V1-18	9343	969.2	10497	788.1	2843	89	27	0.9	−0.23
V2-09	7784	969.5	7392	788.2	1955	105	26	0.6	−0.17
V3-01	9898	967.9	25587	787.0	4987	39	19	2.2	−0.58
V3-09	9002	969.0	14240	787.9	4263	63	30	1.1	−0.29
V3-15	7319	969.2	14205	787.5	3890	52	27	0.9	−0.24
V3-16	7868	968.8	3121	787.5	1274	252	41	1.3	−0.34

## Data Availability

The used measurement data are available on request from the corresponding author.

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
