# Peer review of "Experimental Analysis of Ductile Cutting Regime in Face Milling of Sintered Silicon Carbide"

_materials, 2022, doi:10.3390/ma15072409_

Round 1
Reviewer 1 Report
The article deals with ductile cutting regime in face milling of sintered silicon carbide. The article is of some interest, but the necessary changes and additions must be made before acceptance.
The abstract is too long and contains a lot of unnecessary information. I would recommend deleting the first three phrases and starting right away with the essence of the research.
Figure 1 - how can this information be useful for understanding the article? Was it created by the authors or borrowed (then there should be a corresponding link)?
Figure 2 - I'm not sure if this graph is needed in a scientific article. What implications does this have for understanding the results of the study?
Teutology: "Scanning acoustic microscopy was performed at... SAM" - it is also necessary to give a transcript of the abbreviation XRD, SAM.
Figure 3 in its current form is not very informative. I recommend adding a designation for each essential element (1...2...3...) with a corresponding description in the caption.
Figure 3 and Figure 5 can be combined quite well.
Table 2 - it is necessary to provide an identical number of decimal places for one data series.
Figure 17 - Something is wrong with this image. The lower part should be significantly lighter, now it is not possible to understand what the red arrows are pointing to.
Self-closed cracks should be presented at a significantly higher magnification. Now almost nothing is visible.
The conclusion needs to be revised. It is necessary to present the main results of the work in a focused and clear way, preferably on points 1. ... 2. ... etc.
Author Response
Dear reviewer,
Thank you for your helpful suggestions. The reply is attached.
Please see the attachement.

Reviewer 2 Report
The paper lacks an explicit indication of how acoustic emission signals can be used to optimize the machining modes of superhard materials. Conventional units on the acoustic emission level scale do not allow for other processing conditions to determine the permissible level of acoustic emission, which does not lead to significant cracks in the processed product.
Author Response
Dear reviewer,
The potential use of the AE signals has been explored further in the conclusion.
Thank you for your suggestion in making this a better paper!
As a side note:
A major rewrite of the methods section in chapter 2 was undertaken to avoid similarity detection to another publication written by the main author. Figure 3 and 5 were combined. The abstract was shortened by advise of another reviewer.
Reviewer 3 Report
Dear Authors and Editors,
I highly recommend the paper for publishing, since the experiments were planned and performed in excellent way, and the analysis of results was correct. I only suggest two things to be improved:
- it is recommended to provide some information on the measurement accuracy, which is especially interesting in the case of few dozens of nanometers (Tabs. 2-5),
- the state-of-art review should be updated, adding some more recent positions, since there are only 7 papers covering last 5 years out of total 40.
Author Response
Dear reviewer,
Thank you for your kind words in regard to the paper.
The measurement accuracy of the CLSM was updated in chapter 2 ("Materials and Methods").
Additional, more recent positions have been added to the state-of-the-art review.
As a side note:
A major rewrite of chapter 2 was undertaken to avoid similarity detection to another publication written by the main author. Figure 3 and 5 were combined. The abstract was shortened by advise of another reviewer.
Round 2
Reviewer 1 Report
Since the authors have made the necessary corrections in accordance with the recommendations of the Reviewer, the manuscript can be recommended for publication.